# Experimental Investigation of Adhesion Failure between Waterproof Coatings and Terrace Tiles under Usage Loads

**Barbara Francke and Artur Piekarczuk \***

Instytut Techniki Budowlanej, Filtrowa 1, 00-611 Warsaw, Poland; b.francke@itb.pl
\* Correspondence: a.piekarczuk@itb.pl

**Abstract:** This paper analyses the mechanism of the loss of functional properties of water-impermeable products used under ceramic tiles bonded with adhesives. Recorded damages were caused by selected ageing factors and were measured by the loss of adhesion of individual layers of the set. The analyzed phenomenon is found mainly on terraces and balconies located in a mid-European transitional climate, i.e., exposed to temperatures passing through 0 °C for three seasons a year. The tests reflected the action of three main functional factors, i.e., temperatures, water and freeze/thaw cycles. Tested waterproof coatings were grouped into three types, i.e., dispersion, cementitious and reaction resin-based products. Research kits consisted of liquid-applied water-impermeable products laid on a concrete substrate, adhesives and tiles. Comparing the effects of the action of the above-mentioned ageing factors revealed that water has the greatest impact on the reduction of the tensile adhesion strength of such sets. The adhesion of waterproof coatings to the concrete substrate showed higher values than the adhesion between the waterproof coating and the tile adhesive layers, regardless of the coating material. Both for samples not exposed to ageing factors, and for those exposed to such impacts, failure usually occurred in the adhesive layer or between the tile adhesive and the waterproof coating, without damaging the waterproof layer. The loss of adhesion of finishing layers to the substrate was not accompanied by a loss of tightness of the waterproof coating. The impact of negative water ageing was particularly destructive on the adhesion of cement-based tile adhesives to waterproof coatings made of polymer with a water dispersion of absorbability above 7% (V/V). There was no correlation among the results of adhesion of the finishing layers to the waterproofing layer after the action of the three ageing factors, i.e., water contact, elevated temperature and freeze/thaw cycles.

**Keywords:** liquid-applied waterproof coatings; durability and sustainability; ageing changes

## 1. Introduction

The majority of processes which destroy building materials take place in the presence of water or moisture and that is why structures must be protected from the ingress of unwanted rain water, water accumulated in the soil [1], water splashed on the floor in wet rooms, as well as water which accumulates on the surface of terraces and balconies. This article is devoted to assessing the durability of one of the above-mentioned areas of application of waterproofing, i.e., waterproofing layers of terraces and balconies, assuming that the durability of this part of the covering determines the proper protection of the structure against water and moisture. This area is additionally exposed to temperature variations along daily or seasonable cycles, inducing expansions and contractions in building materials [2]. Depending on the type of material and extent of such temperature variations, over the years the material can fatigue and may result in damage by the onset of brittle failure. In mid-European transitional climates, terraces and balconies are exposed to temperatures passing

through 0 °C for three seasons in a year. The surface temperature of a tile depends on a number of factors related to illumination; the environment in case of weathering, temperature and intensity of rain, hail or snow, the physical properties of the tile, and the physical properties of the substrate materials [2]. Various studies have investigated the link between thermal variations and material failure for different building materials. In particular, the shrinkage and expansion of cementitious materials like tile adhesive mortars, in relation to the presence of water and temperature changes, have been discussed in many scientific articles. In those investigations, the tile–mortar interface was found to be a zone, where failures often occur first. Microstructures of polymer-modified tile adhesives have been investigated with respect to adhesion properties [3–9]. Numerous analyses have also been carried out in terms of tensile and shear stresses appearing in the structure of the adhesive mortar, depending on its composition, and the effect of tile size and colour on the adhesion of ceramic tiles to the substrate. For example, R. Zurbriggen and M. Herwegh [2] investigated ceramic tiles laid on different substrates depending on thermal actions (e.g., slow heating of the entire structure in the afternoon, or rapid cooling of the tiles in a rainstorm). They performed a field experiment with real outdoor tilings over a period of three years, including the real-time monitoring of thermal and failure evolution. They found that over the years, ongoing cycles of thermal expansion and contraction result in material fatigue, promoting the propagation of cracks. Test results, i.e., material properties and geometric aspects of the real tilings, formed the basis for the creation of numerical modelling. The authors suggested how to counteract the generation of critical stresses in outdoor applied ceramic tilings. Studies on large-sized tiles performed by Wetzel et al. [10] show that the adhesion properties of a tile adhesive vary in respect to the position (centre vs. rim). The rim of tiles was found to be the most critical position with respect to failure evolution. As a consequence, such spatial variations in strength require spatially resolved microstructural investigations. Along the aforementioned critical position at the rim of the tile, primary drying as well as potential rewetting by water intake takes place. Both drying and rewetting are coupled to a volume change, which may lead to local stress accumulations. The length changes of substrates and the surfaces of single applied tiles have been measured in previous studies under wet, dry and alternating storage conditions. Wetzel et al. [10,11] studied the interplay between water infiltration and cracking, beginning at the early curing of the mortar during the first days after application until water transport in the hardened system, and the impact of a waterproofing membrane (applied between substrate and tile adhesive) on the behavior of ceramic tiles. They found that the waterproofing membrane reduces crack propagation. Herwegh et al. [12] identified potential sites of material failure in the tile–mortar–substrate systems, and investigated the locations and intensities of stress concentrations owing to drying-induced shrinkage. They also confirmed that the flexible waterproofing membrane could strongly reduce stresses in the adhesive mortar in the case of substrate shrinkage. This membrane blocks water migration across the tiling system and has the ability to over-bridge cracks formed in the substrate. Due to its low elastic modulus, it acts as a deformable layer between the rigid substrate and tiles. In the above-mentioned technical reports, the waterproofing layer was usually classified in terms of its watertightness, without taking into account the possibility of using materials of different chemical compositions. In our research, we tried to determine how various waterproofing products affect the adhesion of the set: concrete substrate, waterproofing, tile adhesive, ceramic tile, in conditions of changing temperatures, i.e., under and over 0 °C, in the presence of water. Our research reflects the action of three main functional factors, i.e., temperatures in the (+70 ± 3) °C range, water at (23 ± 2) °C, freeze/thaw cycles at temperatures ranging from (−15 to ±3) °C to (+15 ± 3) °C. We also attempted to find out if a loss of the adhesion of the finishing layer affects the watertightness of the waterproofing layer of terrace covers. In addition, an attempt was made to establish a correlation between values of adhesion after various ageing factors. Such findings would allow one to draw inferences regarding the behavior of the set in different conditions of impact based on the assessment of the results obtained for one selected ageing load.

As the subject of the article is the assessment of the durability of various waterproofing systems, it was assumed that the tested systems used the same type of substrate and the same type of ceramic tiles,

while the waterproof coating products varied. For research purposes, coating products were grouped into three types, in accordance with the classification [13,14] given in the harmonized standard EN 14891 [14], i.e., dispersion (DM, dispersion liquid-applied water-impermeable products), cementitious (CM, cementitious liquid-applied water-impermeable products) and reaction resin-based (RM, reaction resin liquid-applied water-impermeable products) [15].

## 2. Materials and Methods

### 2.1. Materials

Based on initial elimination tests performed on different waterproof coverings in the same three groups, i.e., dispersion liquid-applied products, reaction resins and cementitious mortars, we chose eight materials possessing technical characteristics, which are common for the represented material groups. Four (4) separate sample batches were prepared from each tested set, i.e., one sample batch for simulations of each impact and one reference batch, intended for initial adhesive strength assessment. Each of the test sets was composed of the following layers (in order of arrangement):

- test substrate, i.e., concrete substrate [16], with dimensions 50 × 250 × 550 mm, made of Portland cement CEM I 42,5R, sand-gravel mix with grain size from 0 mm, ratio of binder to aggregate in the mixture 1:5 (by mass), W/C 0.5 to 8 mm, with a roughness indicator of about 0.46 mm and humidity about 2%,
- waterproof coating, applied as below, made of liquid-applied waterproofing product. The product sets were marked from 1 to 8 for the purpose of the test, and their characteristics are presented in Table 1,
- layers of the adhesive [17], the best for use with a specific waterproof coating (available as part of a system cooperating with a specific coating, different for each coating). Basic information about the applied adhesive layer is given in Table 1,
- nine ceramic tiles of V1 type [18], with water absorption ≤0.5% by mass, unglazed, with plain adhering surface with facial dimensions of (50 ± 1) mm × (50 ± 1) mm.

**Table 1.** Short characteristics of the products used in test sets.

| Test Set Number | Type of Waterproof Coating | Type and Grade of Adhesive Mortar | Watertightness at 150 Kpa for 7 Days | Water Absorption of Coating after 7 Days | |
|---|---|---|---|---|---|
| | | | | Weight Increase, G | V/V, %, |
| 1 | DM [1] | C2S1 | No leakage | 6.33 | 8.05 |
| 2 | RM [2] | C2 | No leakage | 0.15 | 0.32 |
| 3 | RM [2] | C2 | No leakage | 0.16 | 0.34 |
| 4 | RM [2] | C2 | No leakage | 0.18 | 0.29 |
| 5 | DM [1] | C2TE | No leakage | 0.55 | 1.75 |
| 6 | CM [3] | C2TE | No leakage | 4.98 | 19.82 |
| 7 | DM [1] | C2TS1 | No leakage | 14.00 | 43.69 |
| 8 | DM [1] | C2TE | No leakage | 1.00 | 7.3 |

[1] waterproof coating made of dispersion liquid-applied products. [2] waterproof coating made of reaction resins. [3] waterproof coating made of cementitious mortars.

The test area of each sample enabled the application of at least nine metal pull head plates with thicknesses of 10mm, 50 × 50 mm each, at a distance of at least 50 mm from one another, so that the mean value from nine measurements was the test result. Every metal plate had a suitable fitting for connection to the test machine.

Tested products can be characterised as follows:

- CM—cementitious liquid-applied water-impermeable products (so called cementitious thin-layer waterproofing mortars). These polymer-cement mortars include: cement, selected mineral

aggregate with a grain size selected according to a specially developed screening curve, fibres and specific additives (specially modified resins, hydrophobic compounds etc.). This composition guarantees an effective waterproofing effect, even with small layer thicknesses. Added to this is a water polymer dispersion (or redispersible copolymers) that provides significant flexibility of the mortar after drying. An additional advantage of cement mortars is their ability to be applied on wet substrates. These mortars bind by hydration and drying.

- DM—liquid-applied water-impermeable products. They are solvent-free, consisting of the water dispersion of polymers. They guarantee full moisture protection and surface covers, even at layer thicknesses of 1.0 mm. They are characterised by good adhesion to various substrates and considerable flexibility. They dry by evaporating water.
- RM—reaction resin liquid-applied water-impermeable products. These are one- or two-component, solvent-free resins, consisting of synthetic resin components (usually based on polyurethanes), with the addition of fillers, pigments and modifiers. They provide substrate protection and watertightness when exposed to moisture and water in the presence of aggressive media. They are characterised by flexibility and very good adhesion to the substrate.

All coatings for test samples consisted of two layers. Sets n° 4 and 6 were made of two-component products and sets n° 1, 2, 3, 5, 7 and 8 were made of one-component products. In the case of two-component products, the ingredients were thoroughly mixed together before applying the cover, and consisted of:

- set no 4: two reaction resin liquid components, no primer before application, total thickness of dried membrane was 2.0 mm,
- set no 6: two components: the first was a liquid emulsion and the second a cementitious powdery mortar, no primer before application only wetting the substrate with water, total thickness of dried membrane was 0.8 mm.

In both cases, the entire packaging of both components was mixed together.
Sets n° 1, 2, 3, 5, 7 and 8 consisted of:

- set no1: dispersion liquid component ready for use after thorough mixing, substrate primed with the same chemical base, total thickness of dried membrane was 2.5 mm,
- set no 2: reaction resins liquid component ready for use after thorough mixing, no primer before application, total thickness of dried membrane was 1.5 mm,
- set no 3: reaction resins liquid component ready for use after thorough mixing, no primer before application, total thickness of dried membrane was 1.5 mm,
- set no 5: dispersion liquid component ready for use after thorough mixing, substrate primed with the same chemical base, total thickness of dried membrane was 2.0 mm,
- set no 7: dispersion liquid component ready for use after thorough mixing, substrate primed with the same chemical base, total thickness of dried membrane was 1.0 mm,
- set no 8: dispersion liquid component ready for use after thorough mixing, substrate primed with the same chemical base, total thickness of dried membrane was 2.0 mm.

The thickness of coatings adopted in the tests was based on the manufacturer's recommendations as the optimal value for transferring comparable service loads, which was the subject of the tests.

*2.2. Methods of Tests*

As the purpose of the research was to reflect the action of three main functional factors, i.e., temperatures in the $(+70 \pm 3)$ °C range, water at $(23 \pm 2)$ °C, freeze/thaw cycles at temperatures ranging from $(-15 \pm 3)$ °C to $(+15 \pm 3)$ °C, our samples were exposed to the following ageing factors [14]:

- thermal impact at temperatures $(+70 \pm 3)$ °C,

- water impact at temperatures (23 ± 2) °C,
- freeze/thaw cycles at temperatures ranging from (−15 ± 3) °C to (+15 ± 3) °C.

The initial adhesive strength value of the abovementioned systems was taken as the basic characteristic and used as a comparative value for the assessment of ageing changes, following the simulation of operating impacts under laboratory conditions. In order to determine whether the ageing factors cause a loss of watertightness, the chosen samples were additionally subjected to a water resistance test.

### 2.2.1. Initial Tensile Adhesion Strength

The tests began with an initial tensile adhesion strength assessment under laboratory conditions, i.e., at (23 ± 2) °C and RH (50 ± 5)%. No less than 24 h after application of the coating, ceramic tiles of type V1 were glued and, after that, pressed for at least 30 s with (20 ± 0.05) N load. The samples were conditioned for 28 days under laboratory conditions, i.e., at (23 ± 2) °C and RH (50 ± 5)%. In the meantime, the pull head plates were bonded before the end of seasoning, i.e., after 27 days. The test was conducted as follows [14]:

- the samples were cut through to the surface of the concrete slab, around the perimeter of each tile,
- the tensile adhesion strength was determined by applying a force at a constant rate of (250 ± 50) N/s, perpendicularly to the coating surface (without any bending stress against the axis of the currently tested pull head plates),
- the maximum force at which the pull head plate was torn off was recorded,
- the mean force was calculated from nine measurements, in N.

The coating adhesion test stand is presented in Figure 1.

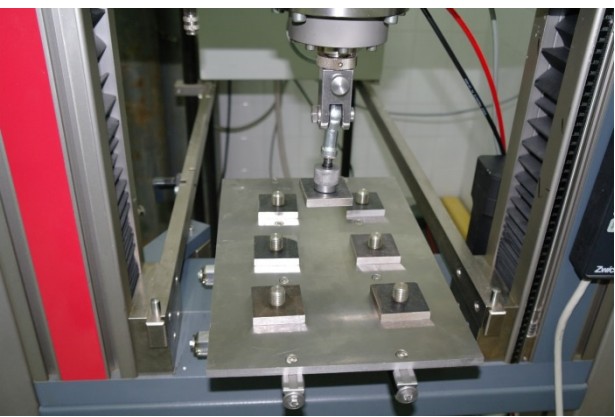

**Figure 1.** Coating adhesion strength test stand.

### 2.2.2. Resistance to Elevated Temperatures

Resistance to elevated temperatures was tested at a temperature of (+70 ± 3) °C.

No less than 24 h after application of the coating, ceramic tiles of type V1 were glued and after that pressed for at least 30s with (20 ± 0.05) N load, the same as for the initial tensile adhesion strength test. The samples were conditioned for 14 days under laboratory conditions, i.e., at (23 ± 2) °C and RH (50 ± 5)%. The test was conducted as follows [14]:

- the samples were conditioned for 14 days in a dry oven at temperatures of (70 ± 3) °C,
- pull head plates were bonded after 14 days,
- the samples were cut through to the surface of the concrete slab, around the perimeter of each tile,

- the tensile adhesion strength was determined by applying a force at a constant rate of (250 ± 50) N/s, perpendicularly to the coating surface (without any bending stress against the axis of the currently tested pull head plates),
- the maximum force at which the pull head plate was torn off was recorded,
- the mean force was calculated from nine measurements, in N.

The watertightness test was performed for samples in which the detachment of the metal pull head plates took place in the finishing layers. The water pressure at 150 kPa acted from above the specimen for seven days.

### 2.2.3. Resistance to Water

Resistance to water was tested in tap water at a temperature of (23 ± 2) °C (Figure 2). No less than 24 h after the application of the coating, ceramic tiles of type V1 were glued and, after that, pressed for at least 30 s with (20 ± 0.05) N load, the same as for the initial tensile adhesion strength test. The samples were additionally protected on all side surfaces, and at the bottom surface with waterproof coating. The test was conducted as follows [14]:

- the samples were conditioned for seven days under laboratory conditions, i.e., at (23 ± 2) °C and RH (50 ± 5)%,
- after that, they were immersed in water with standard temperature for 20 days,
- after the above period, the samples were removed from the tank, wiped with a cloth, and pull head plates were glued to the tested coat,
- next, after seven hours, the samples were re-immersed in water in standard temperature for 24 h,
- an adhesive strength test was performed immediately after removing samples from the water, the same method as for the initial tensile adhesion strength,
- the maximum force at which the pull head plate was torn off was recorded,
- the mean force was calculated from nine measurements, in N.

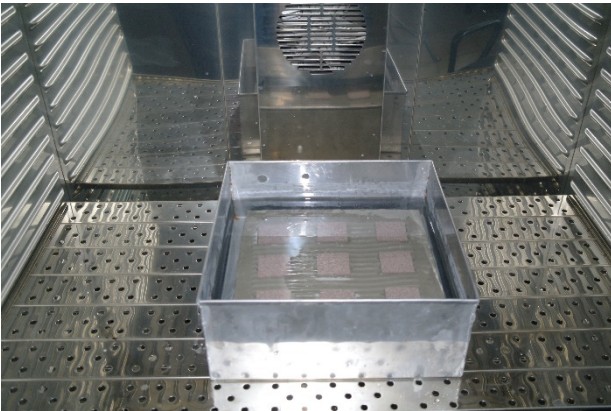

**Figure 2.** View of the sample during water resistance testing at (23 ± 2) °C.

The watertightness test was performed for samples, in which the detachment of the metal pull head plates took place in finishing layers. Water pressure at 150 kPa acted from above the specimen for seven days.

### 2.3. Resistance to Freeze/thaw Cycles

Resistance to freeze/thaw cycles was tested under variable temperatures, ranging from (−15 ± 3) °C to (+15 ± 3) °C. No less than 24 h after the application of the coating, ceramic tiles of type V1 were glued and, after that, pressed for at least 30s with (20 ± 0.05) N load, the same as for the initial tensile

adhesion strength test. The samples were additionally protected on all side surfaces and the bottom surface with waterproof coating. The test was conducted as follows [14]:

- the samples were conditioned for seven days under laboratory conditions, i.e., at $(23 \pm 2)$ °C and RH $(50 \pm 5)$%,
- after that, they were soaked in water at standard temperature for 21 days,
- next, 25 freeze/thaw cycles were carried out: maintaining the test pieces at temperature $(-15 \pm 3)$ °C for 2 h $\pm$ 20 °C after removing the samples from water. Time necessary to reduce the temperature: 2 h $\pm$ 20 min,
- immersed in water at $(20 \pm 3)$ °C, raising the temperature to $+(15 \pm 3)$ °C and maintaining this temperature for 2 h with $\pm$20 min tolerance,
- after that, the samples were removed from the water, wiped with a cloth, mounted in testing holders and seasoned under laboratory conditions for at least seven hours,
- an adhesive strength test was conducted, the same method as for the initial tensile adhesion strength,
- the maximum force at which the pull head plate was torn off was recorded,
- the mean force was calculated from nine measurements, in N.

Numerous earlier tests, carried out in accordance with the above-mentioned cycle, allowed one to state that the reduction of the adhesion of the test system and possible damages to the waterproof coating occur during the first 25 cycles. Accordingly, the number of research cycles used in this study was limited to 25.

The watertightness test was performed for samples, in which the detachment of the metal plates took place in the finishing layers. Water pressure at 150 kPa acted from above the specimen for seven days.

## 3. Results and Discussion

This paragraph presents and discusses the results of the research. The test results of the initial tensile adhesion strength are presented in Table 2. The test results of the adhesion strength after exposure to the following ageing factors [14], i.e.,

- thermal impact at temperature $(+70 \pm 3)$ °C,
- water impact at temperature $(23 \pm 2)$ °C,
- freeze/thaw cycles at temperatures ranging from $(-15 \pm 3)$ °C to $(+15 \pm 3)$ °C,

are presented in Table 3. The obtained adhesion values were supplemented with an assessment of the mode of failure and with information about the coating's watertightness after every ageing factor.

**Table 2.** Results of initial adhesive strength for eight coating sets.

| Test Method | Tested Set Number/Symbol of Coating Type/Test Result (Mean Value)—Tensile Adhesion Strength, N/Mm$^2$/Watertightness (Mean Value) | | | | | | | |
| --- | --- | --- | --- | --- | --- | --- | --- | --- |
| | 1 | 2 | 3 | 4 | 5 | 6 | 7 | 8 |
| | DM | RM | RM | RM | DM | CM | DM | DM |
| Initial tensile adhesion strength | 0.60 *) | 0.97 **) | 1.40 **) | 0.36 **) | 1.21 **) | 0.67 *) | 0.62 *) | 1.33 *) |

Mode of failure: *) within tile adhesive layer, **) between adhesive layer and waterproof coating. DM—waterproof coating made of dispersion liquid-applied products, RM—waterproof coating made of reaction resins, CM—waterproof coating made of cementitious mortars.

**Table 3.** Results of adhesive strength after ageing tests for eight coating sets.

| Test Method | Tested Set Number/Symbol of Coating Type/Test Result (Mean Value)—Tensile Adhesion Strength, N/Mm$^2$/Watertightness (Mean Value) | | | | | | | |
|---|---|---|---|---|---|---|---|---|
| | 1 | 2 | 3 | 4 | 5 | 6 | 7 | 8 |
| | DM | RM | RM | RM | DM | CM | DM | DM |
| – after heat ageing | 0.56 *) | 0.81 **) | 1.29 **) | 0.16 **) | 1.08 *) | 0.69 *) | 0.48 **) | 1.62 *) |
| | Watertightness of the waterproof coating at 150 kPa for seven days—no leakage | | | | | | | |
| – after water contact | 0.34 **) | 0.51 **) | 0.48 **) | 0.42 *) | 0.67 **) | 0.67 *) | 0.25 **) | 0.0 **) |
| | Watertightness of the waterproof coating at 150 kPa for seven days—no leakage | | | | | | | |
| – after freeze/thaw cycles | 0.94 *) | 0.94 **) | 0.94 **) | 0.66 **) | 0.76 *) | 0.72 *) | 0.52 *) | 0.0 **) |
| | Watertightness of the waterproof coating at 150 kPa for seven days—no leakage | | | | | | | |

Mode of failure: *) within tile adhesive layer, **) between adhesive layer and waterproof coating. DM—waterproof coating made of dispersion liquid-applied products, RM—waterproof coating made of reaction resins, CM—waterproof coating made of cementitious mortars.

Tests have been carried out for the following sets marked with numbers from 1 to 8, in which the waterproofing layers were made of covers from:

- dispersion liquid—applied products, four sets, marked with the symbol DM and numbers: 1, 5, 7 and 8,
- reaction resins—three sets, marked with the symbol DM and numbers: 2, 3, 4,
- cementitious mortars—one set, marked with the symbol CM and number 6.

The tensile adhesion strength values in Tables 2 and 3 are mean values calculated on the basis of nine measurements.

In all tested cases, the loss of adhesion after ageing factors mainly occurred at the border waterproof coating, namely the tile adhesive or in the tile adhesive layer. In no case was the waterproof coating damaged, which indicates that the weakest point of the tested sets was the adhesive layer, from an adhesion point of view. The adhesion between waterproof coatings and concrete substrates was definitely higher than the adhesion between waterproofing layers and tile adhesives. Water is the main factor to which waterproofing membranes are exposed, and building structures should be protected from it by membranes. The tensile adhesion strength loss for this exposure can reach 100% versus the original value. Generally, it is not the waterproof coating which is damaged, but adhesion reduction occurs in the finishing layers, i.e., in the tile adhesive layer, which may cause gradual detachment of the finishing layer under functional conditions. Tests of tensile adhesion strength following exposure to water were performed immediately after the samples were removed from a liquid medium, i.e., in reference to wet layers, because this reflects the actual conditions of use. In real life, it is not possible to ban entrance to a terrace/balcony after precipitation, in order to wait for the adhesive layer to dry, so as not to expose the wet coating to mechanical loads.

A positive phenomenon was the fact that the waterproof coating had the same level of water resistance after the action of ageing factors, as that found in new coating. This means that after the ageing factors described in item 2 of this article, the coating still protected the terrace/balcony structure against the ingress of rainwater, despite the loosening of adherence of the finishing layers.

According to the approved technical requirements [14], it was assumed that the strength of the set adhesion to the substrate, both initially and following exposure to ageing factors, should not be less than 0.5 N/mm$^2$. The graphic comparison of the obtained values with the limit value is shown in Figure 3. The limit value is marked with a blue horizontal line.

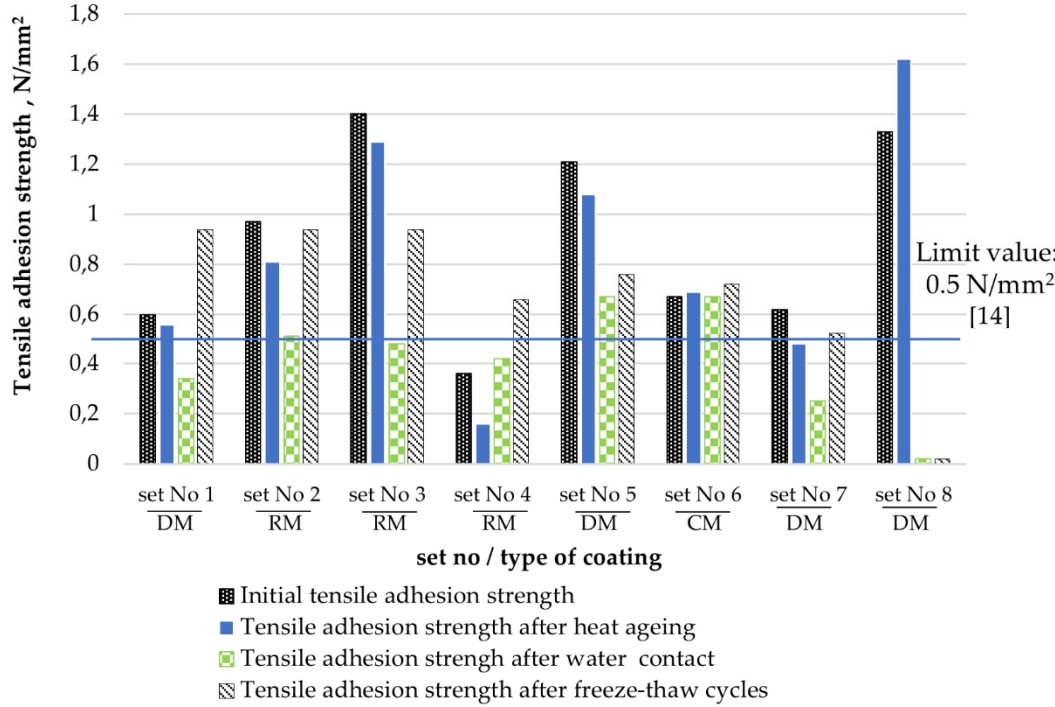

**Figure 3.** Comparison of initial tensile adhesion strength of waterproofing sets, following exposure to selected ageing factors.

It can be seen that, for eight random test sets, only three meet the requirement, which qualifies them for further analyses. The fourth option marked with "set 3" meets the abovementioned assumption only towards the measurement of error limit values. Based on the presented results, it can be alleged that nearly half of commercially available under-tile waterproof coatings do not guarantee the transfer of functional loads, at least on a satisfactory level and, unfortunately, the problem is identified on site, when significant detachment is observed or—in an optimum case—after ageing tests. Only in the case of "set 4" was the phenomenon observed on the stage of the initial tests, i.e., when initial tensile adhesion strength was evaluated. For the other three cases with negative evaluation results, the initial tensile adhesion strength was at least satisfactory, and for one option it revealed very high values.

By far the largest loss of adhesion, regardless of the type of waterproof coating used, was found after water treatment. Such a phenomenon has been described before in the technical literature. For example, Wetzel et al. [10] proved that longer storage under wet conditions might lead to a connection of the crack at the tile–grout interface and the tile–mortar interface, enabling water infiltration into the mortar. Water infiltration into the mortar, however, would lead to swelling (up to 0.5 mm/m) [19], generating local expansion. The resulting stress increase might induce the propagation of pre-existing cracks. Additionally, cement hydration can restart, leading to chemical shrinkage. This creates additional internal porosity, refines the pore structure and may create additional capillary tension during wetting and drying processes. This may increase shrinkage stresses, which can widen pre-existing cracks and even cause the formation of new vertical cracks.

An attempt was made to establish a relationship between the initial tensile adhesion strength value and the adhesion strength value after exposure to ageing factors. At the first stage of the study, a percent change in the adhesion strength of the sets was calculated following exposure to ageing factors, in reference to the initial values, using equation:

$$z = |(R_a - R_i)/R_i| \times 100\% \tag{1}$$

where:

z—change of tensile adhesion strength after ageing factors
$R_a$—tensile adhesion strength after ageing factors
$R_i$—initial adhesion strength

The results are presented in Table 4. The results obtained for the test sets, which did not meet the functional requirements assumed for adhesion strength at min 0.5 N/mm² [14], are given in italics, while the values in bold are the ones which can conditionally be regarded as meeting the requirements, but are close to the measurement error. The range of changes in the adhesion strength of different test sets is presented in graphic form in Figure 4.

**Table 4.** Comparison of percent change in the sets' adhesion strength after exposure to ageing factors versus the initial value.

| Characteristics | Tested Set No./Type of Coating | | | | | | | |
|---|---|---|---|---|---|---|---|---|
| | 1 | 2 | 3 | 4 | 5 | 6 | 7 | 8 |
| | DM | RM | RM | RM | DM | CM | DM | DM |
| Initial tensile adhesion strength, N/mm² | 0.60 | 0.97 | 1.40 | 0.36 | 1.21 | 0.67 | 0.62 | 1.33 |
| Tensile adhesion strength change after thermal ageing, % | −6.7 | −16.5 | −7.9 | −55.6 | −10.7 | −3.0 | −22.6 | +21.8 |
| Tensile adhesion strength change after water impact, % | −43.3 | −47.4 | −65.7 | +16.7 | −44.6 | 0.0 | −59.7 | −100.0 |
| Tensile adhesion strength change after freeze/thaw cycles, % | +56.6 | −3.1 | −32.9 | +83.3 | −37.2 | +7.5 | −16.1 | −100.0 |

DM—waterproof coating made of dispersion liquid-applied products, RM—waterproof coating made of reaction resins, CM—waterproof coating made of cementitious mortars.

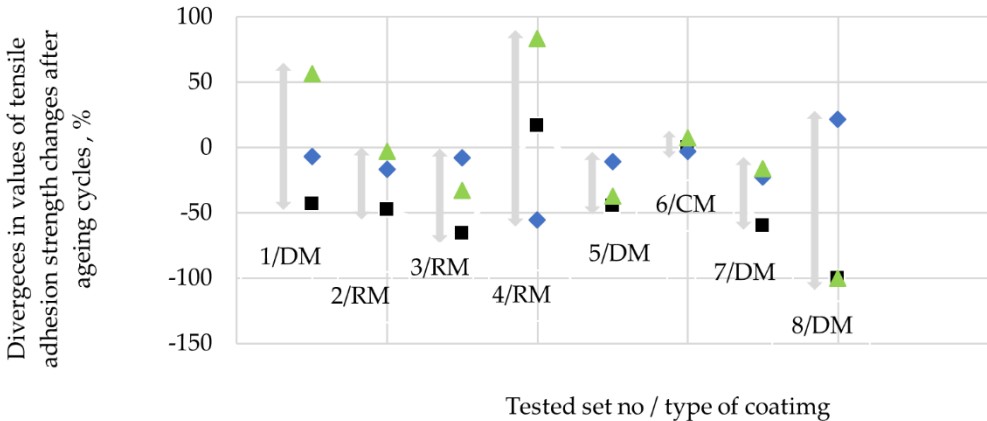

**Figure 4.** Divergences between extreme change values of tensile adhesion strength changes for waterproofing sets, following exposure to ageing factors, in reference to the initial tensile adhesion strength.

Divergences in the values of tensile adhesion strength changes for waterproofing sets, following exposure to ageing factors in reference to the initial tensile adhesion strength, are presented in graphic form in Figure 4.

Typical damages as a result of the tensile adhesion strength test are presented in Figure 5. The largest damages are after water impact at the temperature (23 ± 2) °C. Adhesive damages form up to 80% of the tested area and are found over the surface of the waterproof coating. The surface area of adhesive damages after freeze/thaw cycles is almost comparable to the damages caused by water. These damages are up to 60% of the tested area and also locally reach the surface of the waterproof coating. The type and extent of adhesive damages illustrate the moisture distribution in

the samples' cross-section. In the area with the highest moisture, adhesive delamination and a loss of internal adhesion occurs. Thermal ageing does not cause much damage, and detachment occurs in the surface zone in 10% of the test surface. The appearance of the surface after such a test is similar to the appearance of the surface after the initial tensile adhesion strength test.

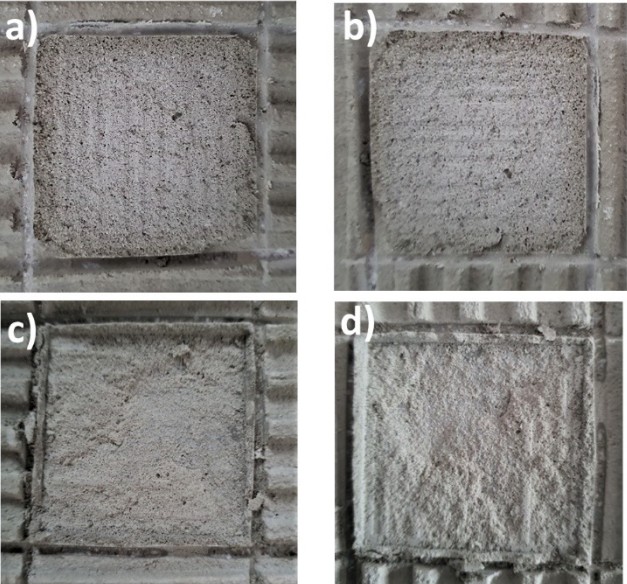

**Figure 5.** Appearance of adhesion surface (after tearing off the ceramic tile) in tensile adhesion strength test: (**a**) initial test, (**b**) after thermal impact, (**c**) after water impact, (**d**) after freeze/thaw cycles.

In reference to all previously obtained test results, we attempted to identify the relationship between the tensile adhesion strength value after exposure to water and the tensile adhesion strength after freeze/thaw cycles. For this reason, we calculated the quotient of these two values, i.e., the value of tensile adhesion strength after water contact, and the value of tensile adhesion strength after freeze/thaw cycles. It was observed that the value never exceeded one. The only exception to the rule was when the tensile adhesion strength following freeze/thaw cycles is an indeterminable value and the set or its fragments detach completely and in an uncontrolled way from the substrate. It can be presumed that when the value is close to one, the tested set becomes more durable, on the condition that all tensile adhesion strength values obtained in the tests meet the assumed requirement, i.e., they are not lower than 0.5 N/mm$^2$. Figure 6 presents the relationship graphically.

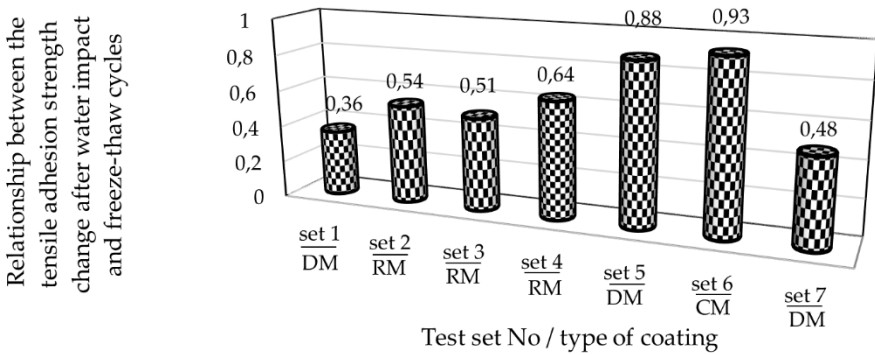

**Figure 6.** Relationship between the tensile adhesion strength change in the set, following water impact and freeze/thaw cycles, determined as a quotient of these two values.

In another stage of the assessment, an attempt was made to compare the tensile adhesion strength following the exposure to ageing factors with the water absorption of waterproof coatings used in the tested set arrangements. The comparison is presented in Table 5 and in Figures 7 and 8.

**Table 5.** Comparison of water absorption of waterproofing membranes with the values of tensile adhesion strength change obtained, following the exposure to ageing factors for eight coating sets.

| Tested Characteristic | Test Set No/Type of Coating/Test Result (Mean Value)—Adhesive Strength N/mm² | | | | | | | |
|---|---|---|---|---|---|---|---|---|
| | 1 | 2 | 3 | 4 | 5 | 6 | 7 | 8 |
| | DM | RM | RM | RM | DM | CM | DM | DM |
| Water absorption of coating after 7 days, weight increase, g/V/V, % | 6.33/8.05 | 0.15/0.32 | 0.16/0.34 | 0.18/0.29 | 0.55/1.75 | 4.98/19.82 | 14.0/43.69 | 1.0/7.3 |
| Change in tensile adhesion strength after water contact, % | −43.3 | −47.4 | −65.7 | +16.7 | −44.6 | 0.0 | −59.7 | −100.0 |
| Change in tensile adhesion strength after freeze/thaw cycles, % | +56.6 | −3.1 | −32.9 | +83.3 | −37.2 | +7.5 | −16.1 | −100.0 |

DM—waterproof coating made of dispersion liquid-applied products, RM—waterproof coating made of reaction resins, CM—waterproof coating made of cementitious mortars.

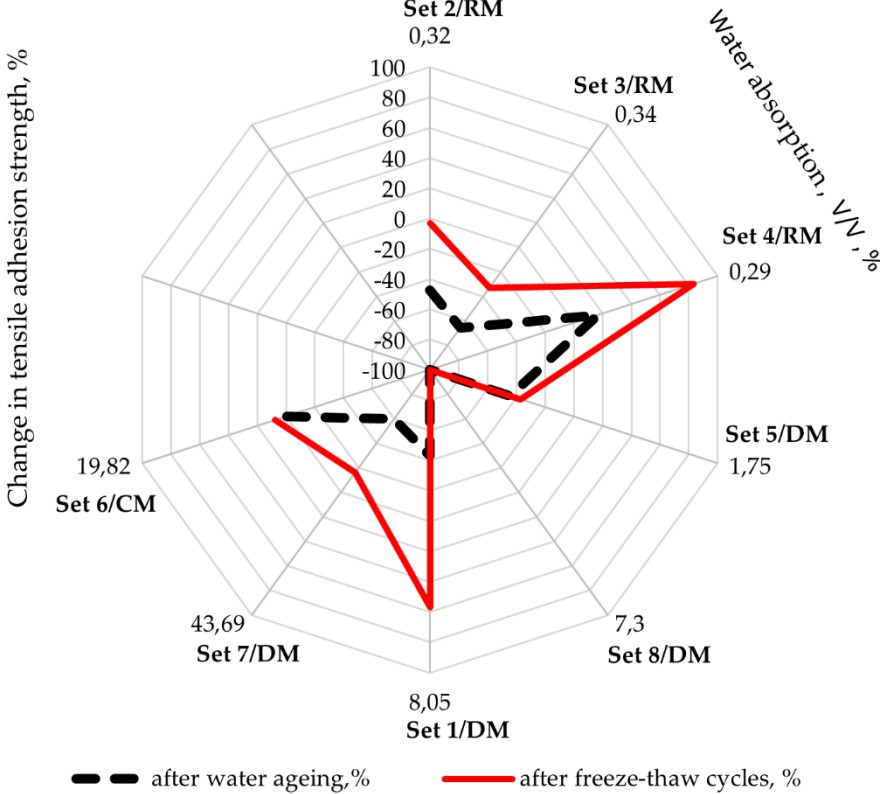

**Figure 7.** Relationship between a change in the test set tensile adhesion strength following water impact and water absorption of coating absorbability, in V/V%. DM—waterproof coating made of dispersion liquid-applied products, RM—waterproof coating made of reaction resins, CM—waterproof coating made of cementitious mortars.

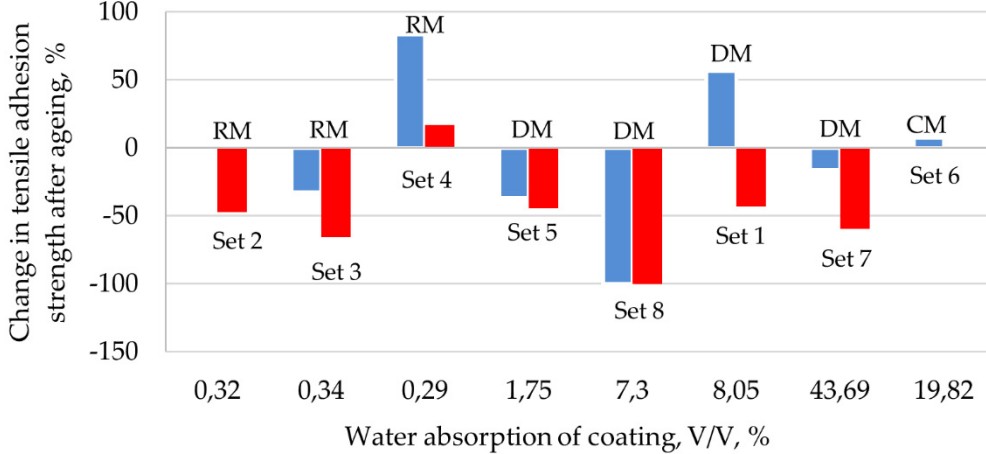

**Figure 8.** Graphic presentation of the change in the test set tensile adhesion strength after freeze/thaw cycles and after water contact, in reference to the water absorption of coatings.

For coatings made of dispersion liquid-applied products, after water ageing, there was a clear relationship between the absorbability of the coating and a decrease in the adhesion between the coating and the tile adhesive, below the permissible limit value, i.e., 0.5 N/mm$^2$ [14]. The coating for which water absorption was of up to 2%(V/V) met the above requirement, while for products with an absorbability above 7%(V/V), the adhesion between the tile's adhesive and the coating was within an unacceptable range. Unfortunately, no correlation was found in this respect for the other two groups of coating products, i.e., for coatings made of reaction resins and cementitious mortars.

The impact of negative water ageing is particularly destructive for waterproof coatings made of polymer water dispersion with organic additives and mineral filler, when they are used together with cement-based tile adhesives. The tensile adhesion strength reduction following exposure exceeds 40%, even if the set can be evaluated positively after ageing, because its tensile adhesion strength still exceeds 0.5 N/mm$^2$. In an extreme case, as a result of destructive water impact, a 100% decrease in the tensile adhesion strength was observed. In all analysed cases, detachment occurred on the border between the waterproof coating and the ceramic tile adhesive. For the presented products, elevated temperatures did not normally cause the destruction of the sets and the drop in the tensile adhesion strength, and compared to the initial value, was no higher than 20%, even in extreme cases (neglecting assessment of the set which did not meet the initial tensile adhesion strength requirement). Freeze/thaw cycles for sets with polymer dispersion waterproof coatings render completely different results, i.e., from a significant increase in the tensile adhesion strength, to a decrease reaching 100%. Following exposure to the ageing factor, it is hard to identify a certain rule pertaining to durability in the tested sets. Generally, when sets composed of waterproof coatings made of polymer dispersion and cement-based tile adhesives do not meet the established requirements, and the divergence of the limit values of tensile adhesion strength changes after exposure to different ageing factors exceeds 40%, values fluctuate mainly around 100%. The divergence for the set made of reference products was much lower, around 30%.

At the current stage of tests, it is hard to formulate significant conclusions for the durability of under-tile waterproof coating sets made of polymer-cement waterproofing products used in a system with cement-based tile adhesive, because only one representative of the group has been tested so far. Nevertheless, a favourable interaction between the coating and the tile adhesive can be expected in this case, due to their chemical compatibility. The initial thesis was confirmed by all results obtained so far for this solution option, where the value of the initial tensile adhesion strength

following all ageing impacts does not deviate significantly from the initial results, and the maximum divergence in the results was 10%. Technical literature [20] pertaining to the products states that water absorption through the reference coatings increases as the P/C (polymer-cement) indicator goes up. Coating-forming accelerating additives reduce its water resistance, but improve its tensile mechanical characteristics [21]. One needs to remember that an excess of the additives is harmful for mechanical properties. The most favourable water-resistant characteristics were demonstrated for coatings with 0.3% of the dispersive additive. A decrease in the P/C indicator contributes to a drop in the value of the coating elongation at break, though the stress value goes up initially to drop significantly after a while [22]. For the P/C indicator value of 0.12, the elongation at break remains at ≥30%.

In Poland, products based on reactive resins are less common for making under-tile waterproof coatings than coatings based on polymer dispersion and polymer-cement products. Three of the studied cases revealed no rules for the distribution of the results, especially since one of the tested sets was disqualified at the preliminary test stage, i.e., tensile adhesion strength testing. In the case of two sets for which the results were positive, all tested ageing factors caused a gradual loss in tensile adhesion strength. Both sets were particularly sensitive to water impact, and once their ageing process had been completed, they met the reference assumptions within a measurement error range. Detachment occurred on the waterproof coating-adhesive coating border. In both cases, the divergence between the limit values of changes in tensile adhesion strength after exposure to different factors was over 50% and even up to 70%.

Detachment of ceramic tiles is often observed in building practice on terraces and balconies, and it rarely occurs in wet rooms. Taking the above into account, it can be alleged that water impact is one of the major factors causing adhesive strength loss. This applies to rain water and water used for surface cleaning, but other ageing factors, including exposure to variable positive and negative temperatures, accelerate destruction processes even further. Water is the main factor to which waterproofing membranes are exposed, and the building structure should be protected from it by membranes. Despite the fact that the impact of elevated temperature contributes less than water to a change/reduction in waterproof coating adhesion to the substrate, one should remember that, in practice, both factors exert a simultaneous impact. Hence, the possibility of accumulated effects of the negative impacts on the structure's behavior exists.

## 4. Conclusions

Ageing laboratory tests, in terms of: water contact at (23 ± 2) °C, a high temperature of (+70 ± 3) °C and freeze/thaw cycles under variable temperatures ranging from (−15 ± 3) °C to (+15 ± 3) °C, of systems intended for use on terraces and balconies, which consist of: concrete substrate, waterproof coating made of liquid-applied water-impermeable products, tile adhesive and ceramic tile, helped to draw the following conclusions:

- comparing the effect of water, elevated temperature and freeze/thaw cycles on terrace sets, in which the waterproofing layer is made of a coating product, and the finishing layer of ceramic tiles is glued with adhesives, it can be stated that water has the greatest impact on the reduction of the tensile adhesion strength to the substrate of liquid-applied products under tiles,
- the adhesion of waterproof coatings to the concrete substrate shows higher values than the adhesion between the waterproof coating and the tile adhesive layer, regardless of the coating material. Both for samples not exposed to ageing factors, and for those exposed to such impacts, failure usually occurs in the adhesive layer or between tile adhesive and waterproof coating, without damaging the waterproofing layer. Accordingly, loss of adhesion of finishing layers to the substrate is not accompanied by loss of tightness of the waterproof coating,
- the impact of negative water ageing is particularly destructive on the adhesion of cement-based tile adhesives to waterproof coatings made of polymer water dispersion with organic additives and mineral filler,

- terrace coverings with coatings made of polymer water dispersion with absorbability above 7%(V/V) are exposed to damage, as a result of a loss of adhesion of the finishing layers to the waterproofing layer,
- there is no correlation between the absorbability of coatings made of reaction resins and cementitious mortars and the adhesion of ceramic tile adhesives after the above-mentioned ageing factors,
- a special case are polymer-cement coatings, in which even high absorbability values (about 20%V/V) do not cause a decrease in adhesion after the above-mentioned ageing, between the coating and the ceramic tile finishing layers. For this reason, it seems that such coatings may be the best products for use in terrace systems in central European transitional climates,
- there is no correlation among the results of adhesion of finishing layers to the waterproofing layer after the action of three ageing factors, i.e., water contact, elevated temperature and freeze/thaw cycles.

**Author Contributions:** Conceptualization, B.F.; methodology, B.F.; software, B.F; validation, B.F.; formal analysis, B.F.; investigation, B.F.; resources, B.F.; data curation, B.F.; writing—original draft preparation, B.F.; writing—review and editing, A.P.; visualization, A.P.; supervision, A.P.; All authors have read and agreed to the published version of the manuscript.

**Funding:** This research received no external funding.

**Conflicts of Interest:** The authors declare no conflict of interest.

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
