# Peer review of "Experimental Investigation of Adhesion Failure between Waterproof Coatings and Terrace Tiles under Usage Loads"

_buildings, doi:10.3390/buildings10030059_

Round 1
Reviewer 1 Report
Well written paper. Despite the fact that I did not find any failures in language or spelling I would recommmend a careful proofreading of the text.
Author Response
Dear Reviewer,
The authors would like to thank for your useful comments, which have been taken into account in order to improve the manuscript. According to the reviewer's recommendations, the manuscript has been carefully proofread
Reviewer 2 Report
The authors presented their work, which is of significance for the field it relates to, on making waterproofing coating with materials of different chemical composition. It could be interesting to the readers in this field. However, major revision is needed before consideration for publishing.
Questions and recommendations:
- Could you provide more information to support the novelty of this study?
- Is there any potential relationship between water tightness and chemical component of waterproofing layer?
- What are the material compositions of DC, DM and RM? What are the molecular structures of the main components of them?
- Are there more descriptions available from the authors or literatures regarding how polymer groups interact with water? With these as the background information, it is easier to explain how the chemistry of waterproof coating affects its water absorption and aging properties.
- More information about how water infiltrates in the multilayer system will help readers to better understand the process of aging and failure. It is recommended to use drawing(s)/picture(s) to show how water penetrates and the resulting situation of such penetration.
- It is recommended to state the key point at the beginning of each paragraph and give detailed discussion afterwards. Some of the sentences are too long, e.g., the ones in Lines 6-8, 12-15 and 446-449 etc. Extensive editing of English language and style is required and typos need to be eliminated by careful proofreading.
- It is suggested that descriptions of experimental results be moved to Results and Discussion.
Author Response
Dear Reviewer,
We would like to thank the reviewers for going through the manuscript and providing valuable suggestions for its improvement. Thanks to their constructive comments, we are able to present a clearer and better revised version than the original manuscript. All the comments have been carefully considered and the current manuscript is substantially improved. The amendments that have been made according to the reviewers’ suggestions are listed in our detailed response. With many thanks for your efforts and best regards,
Point 1: Could you provide more information to support the novelty of this study?
Response 1: We tried to explain the novelty of our study in lines 73-82, i.e. In our research, we tried to determine how various waterproofing products affect to the adhesion of the set: concrete substrate - waterproofing - tile adhesive - ceramic tile in the conditions of changing temperatures i.e. under and over 0oC, in the presence of water. We also tried to find out if the loss of the adhesion of the finishing layer affects the watertightness of the waterproofing layer of terrace covers. In addition, an attempt was made to establish a correlation between values of adhesion after various aging factors.
Point 2: Is there any potential relationship between water tightness and chemical component of waterproofing layer?
Response 2: The exact chemical composition of the products used in the research are the subject of the manufacturer's patents protection. Research products were bought from the market and their selection was based on a comparative assessment of their performance characteristics in each of the three assortment groups. For this reason, at the current stage of research it is difficult to perform a comparative analysis between their chemical composition and product watertightness. Only in the next stages of research we want to try to assess the chemical composition, which would allow inference in the field of activity. Thank you for this useful suggestion
Point 3 What are the material compositions of DC, DM and RM? What are the molecular structures of the main components of them?
Response 3:The only information about material compositions of DC, DM and RM, at the current stage of research are general data, i.e.
- CM – cementitious liquid-applied water-impermeable products (so called cementitious thin-layer waterproofing mortars) - these polymer-cement mortars include: cement, selected mineral aggregate with a grain size selected according to a specially developed screening curve, fibers and specific additives (specially modified resins, hydrophobic compounds, etc.).
- DM - liquid-applied water-impermeable products - they are solvent-free consisting of water dispersion of polimers.
- RM - reaction resin liquid-applied water-impermeable products - these are one or two-component, solvent-free resins, consisting of synthetic resin components (usually based on polyurethanes), with the addition of fillers, pigments and modifiers.
We are currently trying to find out funds for further studies connecting with molecular structure of main components of used products. We hope that the results of these studies we are going to be able to write in subsequent articles
Point 4: Are there more descriptions available from the authors or literatures regarding how polymer groups interact with water? With these as the background information, it is easier to explain how the chemistry of waterproof coating affects its water absorption and aging properties.
Response 4: At present, this information is not available Thank you for this useful suggestion. We will take this suggestion into account in future work
Point 5: More information about how water infiltrates in the multilayer system will help readers to better understand the process of aging and failure. It is recommended to use drawing(s)/picture(s) to show how water penetrates and the resulting situation of such penetration.
Response 5: Figure 5 has been added to the manuscript with the preceding interpretation on lines 281-390. It illustrates , among other things, the effect of water and moisture on the behavior of the adhesive
Point 6: It is recommended to state the key point at the beginning of each paragraph and give detailed discussion afterwards. Some of the sentences are too long, e.g., the ones in Lines 6-8, 12-15 and 446-449 etc. Extensive editing of English language and style is required and typos need to be eliminated by careful proofreading.
Response 6: At the beginning of chapters 2.1, 2.2 and 3 the introduction was added Sentences are shortened especially in lines 6-8, 12-15, 446-449. Additionally, the manuscript has been carefully proofread , according to the reviewer's recommendations.
Point 7: It is suggested that descriptions of experimental results be moved to Results and Discussion.
Response 7: As suggested, we combined two points of the manuscript, i.e. point 3 Results and point 4 Disscusion into one point 3 Results and discussion
Round 2
Reviewer 2 Report
Thanks for answering my questions. I am ok with them. The authors may want to check the format consistency.
Overall conclusion: Minor revision
Author Response
Dear Reviewer,
The authors would like to thank for your useful additional comment, which has been taken into account in order to improve the manuscript. Minor corrections have been made in the introduction, i.e.:
- we add some information about the impact of changes in outside temperatures on the durability of the terrace system,
- two sentences are delete i.e. Tiles laid over waterproofing surfaces on terraces and balconies are only finishing layers and help to drain some of the rainwater flowing to the surface of the terrace ceramic tiles; however, they do not guarantee water or moisture tightness. Interaction between waterproofing layers, substrate and finishing layers is also a major problem.
Also, please find all changes and additions in the previous text highlighted by yellow background.
This manuscript is a resubmission of an earlier submission. The following is a list of the peer review reports and author responses from that submission.
Round 1
Reviewer 1 Report
According to the reviewer, the title of the article is too general. It is unknown from him what scientific news to expect in the article
The Introduction is very poor and in the reviewer's opinion it must be written from the beginning. In general, the authors did not write much about the work in it. Instead, they posted a number of scientifically worthless messages, which discourages further reading of the work.
The reviewer will list some comments below for the introduction, although, knowing that in this form it is unacceptable, the reviewer emphasizes that these are not all comments, but only some of them:
Abstract:
Line 15: You can't say “the temperate climate zone”. This is a bad translation of climate zones. Please use the names of climate zones according to existing classifications. For example, the Köppen climate classification.
Introduction:
Line 19: It cannot be said that “waterproofing membrane is an underestimated component of a building”. On what basis do the authors draw such a conclusion in the first line of the manuscript? It's a common slogan that you can't use in a scientific article. This way of writing, is a confusion of a textbook for students with a newspaper column, not a scientific language, especially for the level of MDPI journals.
Line 25: About 25% of the article reference was cited to the very general opinion that "concrete is among most commonly used substrates under waterproofing coatings and is the most important structural material". This shows how poor the bibliography of the article is, which will be discussed later in the review. Perhaps this is an attempt to cite some authors. There would be nothing wrong with it, if not for the fact that the references are to the general slogan. In a scientific article, you cannot write that "concrete is the most important structural material".
This sentence focused the attention of the reviewer on the bibliography, so before moving on to the next lines of text, he would like to comment on the references.
Because the reviewer does not know the authors of the article (they were not given), you can boldly write about individual references. Numerous works by “Francke B.” are cited in a bad way.
The title cannot be written in English if the original and only title is in a different language. The authors have translated the work title into English, this is not the case. In addition, what does this mean in reference no. 9 "C4"? Other references from Francke B, used in the scared unscientific context, indicate that he / she is one of the authors of the article, citing his work.
Line 25, next sentence: “As the material has been gaining popularity, its users gradually pay more attention to the huge losses caused by its lack of durability”. Why write this? Is this serious? Concrete has been gaining popularity? Users pay more attention to the huge losses caused by its (concrete) lack of durability?
Line 28-39, there is no need to explain to world scientists what it is waterproofing mixture, waterproofing coating etc. The reviewer has the impression that the introduction was written on a different topic than the conducted research. Introduction is not a student handbook. In the following lines, the authors describe topics not relevant to the article. This is not the place to explain how to use waterproofing, etc.
One could comment on errors in individual lines, however the reviewer will comment on the whole fragment, without exceptions:
Line 40-101: This information is completely off-topic. The article is about durability of waterproofing coatings. The reviewer expected here a review of the current state of knowledge on the subject. This is not a new topic, many scientific papers have been published in recent years on this topic. Mentioning them seems necessary to compare the results of the authors with the results of other scientists. Meanwhile, the authors write not only not on the subject, but also not in the scientific standard. We read, for example, sentences about the definition of a balcony, moreover with further unnecessary references. Why authors write:
“A balcony is an external platform available for pedestrian traffic, which is a part of a building structure but does not have a roofing function above any areas.” Do the authors think that the definition of balcony may be an introduction to the topic of their research?
Line 91: Why only references to polymer modified cementitious mortar are given? Why four references for such information? Why for others no?
Line 96: What information is the reference [15] related to?
The reviewer emphasizes once again, the information in lines 1-102 says nothing about durability of waterproofing coatings. The article could start from line 102, but the introduction should contain a comprehensive description of the research on the same topic carried out by others.
Materials and Methods:
Line 124 and line 127: you can't write: „according to the manufacturer's instructions”. Where is the manufacturer listed? Who is the manufacturer? There is no description and the reference.
Minor: line 127 space after [20]
Line 130: Although the authors mention that the V1 type tiles according to [21], the reviewer suggests explaining the V1 type in the text.
Minor: instead Photo 1 shoude be Figure 1. All photos, schemes and charts should be named as Figures.
Line 141, “Scheme 1” Stage 1 is not sufficiently described.
In general, the reviewer considers the explanation of test methods to be inappropriate. Between
linen 140 and 157 there is little text and many diagrams. The diagrams, in turn, contain a lot of illegible text. The reviewer once again has the impression that he was at the presentation and not reading a scientific article. It is necessary to delete Scheme 1, Scheme 2, Scheme 3 and Scheme 4. And describe the methods used in the next sections of the article. There are no explanations or references to the methods in the schemes. The description of freeze-thaw cycles is in the diagram and should be described in the text as a separate sub-chapter. For example, the reviewer first reads about this freeze-thaw method. Why 25 cycles? Where are the references? And why was the chapter written on such a Figure? This state of the material and method description is unacceptable.
Results:
What is the purpose of presenting the same results in the form of a large table and a large illegible chart (Figure 1)? It's the same information. The reviewer suggests replacing the chart type with a chart with points. The horizontal axis should indicate the common feature of the samples tested, quantified and not set number. The presentation of these results is unacceptable. Similarly, in Figure 2 and Figure 3, Figure5.
Discussion:
In general, authors should in this chapter discuss how the results can be interpreted in perspective of previous studies. This is impossible because the authors did not describe other studies on this topic.
Line 226: This information was obvious to the reviewer
Line 246-249: “On the current stage of tests it is hard to formulate significant conclusions as for durability (…)” This is true considering the literature review that the authors carried out. The authors write only about their research. Hence, they sum up that no major conclusions can be drawn.
Line 261: In the introduction, the authors wrote "the temperate climate" (wrong name), and now they talk about Polish climate. Is the climate in Poland a benchmark for "the temperate climate"?
Conclusions:
The conclusions should refer to durability of waterproofing coatings and should contain specific numerical values. The conclusions are descriptive and imprecise.
In summary, the reviewer estimates that the manuscript does not meet the criteria of a scientific article.
Reviewer 2 Report
The manuscript “Durability of waterproofing coatings made of liquid applied water impermeable products” has an interesting objective, but, at the current state need some important revisions. I’m referring in particular to the discussion of the results, which are potentially relevant for the investigation of new waterproofing coatings but are poorly described and discussed. There are a lot of data that are just partially presented.
Figures and tables should be modified in an easier and understandable way. They are often poor described in the text and it makes the manuscript hard to read/understand.
I strongly encourage re-submission after major revisions because the database appears good. Therefore, I see a good chance that a good paper can be made of it.
The manuscript requires a linguistic improvement and reviewed before re-submission.
The following comments can be considered to improve the quality of the publication.
Introduction
The introduction has well written parts and other very hard to follow for the reader in order to keep the objective. In general, I suggest you reduce this paragraph that is too long and reorganize it.
At line 23 you write about “concrete” as one of the most used substrates for waterproofing coatings, but in your abstract you enunciate that your research is on the mechanism between waterproofing coatings and ceramic tile (see line 7). I think you would report a panoramic view about the overall issue concerning different coatings and different substrates but at the current state the way in which it is carried out confuses the reader. I think all the introduction has to be rewrite with the following order (just a suggestion):
- state of art on different coatings (if you are particularly interest about the durability in Poland focus better the problem in this way)
- overall issue (uses: e.g. balcony and terrace…, different substrates and different coatings normally used)
- focus on your research: how your research aims to overpass the issues before explained? Which are the conditions chosen for your tests? Which is the main objective of your research?
Thus, I suggest arguing the whole introductive paragraph with a more logical and more pertinent order.
Materials and Methods
I invite you to rewrite this part: you must describe better the sample preparation and the different types of assemblages. Please, consider the following suggestions.
I suggest you rewrite all this part: you can use schemes to simplify the reading, but the must be more synthetic, and they can’t totally replace the text description. Thus, please, rewrite all:
- more description in the text and less description in the schemes;
- it is very confused: the use of scheme and images should help the reader, so, reorganize them.
- rename all the samples (and maintain the new name into all the text).
- you have to better specify methods and normative used for all the analysis carried out.
Other considerations:
Line 130: you have to briefly illustrate which are the V1 types in ceramic tiles. And you do you know their water absorbability?
Table 1: why do you now talk about adhesive mortar? The water absorption values are previous known? If are results of your text they have to be moved in the results paragraph.
I suggest you add a column in table 1 with a new sample labelling: I don’t know for example DM1, DM2, … RM1, RM2, … or different by the type of adhesive used. It is really hard to follow the current sample identification (V.I. = all tables and figures must have the same sample identification from the begging to the end!).
I think in general it is very hard to follow you in the distinction on mortar, cementitious materials and ceramic materials, please check it in the text and rewrite in a more understandable way.
Line 136: I think it is better calling the named Photo 1 in Figure 1.
Photo 2 is not reported into the text, please reported and change the name in Figure 2.
Scheme 1: which are the manufacture instructions??
Results
The fact that the Table 2 is difficult to understand (for example for * and ** symbology I think it is emblematic of a not clear exposition in the Materials and Methods. It is really hard to understand how your samples are made, and consequently also the results are confused.
Fig. 1: what do you mean for: set No1? It is sample 1/DM after different ageing test and in the initial time? Which are the technical requires (line 167) (are you referring to a specific normative?) which impose the limit value at 0.5 N/mm2? Report them as reference.
Table 2: according to which equation did you estimate the % of adhesion strength? And the divergences in the limit values? You have to write how you assess these values.
Fig. 3: What do you mean for “numerical relationship between the tensile adhesion strength change after water impact and freeze-thaw cycles”??? How did you estimate it?
Fig. 4 many samples have the same name.
In general, in this part you have to increase description of the phenomena verify and emphasize difference when found in order to prepare the material for a good discussion in the next paragraph.
Line 211: “The comparison in Table 4 and Figure 4 and Figure 5” -> none comments about results obtained?
Discussion
This part is strongly poor and lacking. You have to increase it referring your discussion to previous (= 3. Results) tables, figures and values obtained. At the current state it looks more like a conclusion paragraph: here you have to comment your results and you have to use previous literature (from lines 253 to 260) to compare and understand your results not just to report the state of the art.
Thus, finally, you have to totally rewrite the discussion after the new result paragraph reorganization.
ConclusionsI think you can integrate your conclusions recycling some of the argumentations which are now in the discussion. At the same times some of the considerations reported in this paragraph can be suggestions to rewrite the discussion (with the suitable argumentation and referments to the values obtained…).
Thus, you can equilibrate these two paragraphs: discussion has to be increased and more detailed; conclusion can be more synthetic.
If your objective has Poland as region of implementation you can say which is, between the all considered, the best coating or the best assemblage for the Polish climate conditions.